# Eye-tracking of clinician behaviour with explainable AI decision support: a high-fidelity simulation study

**Myura Nagendran** [* 1 2]  **Paul Festor** [* 1 3]  **Matthieu Komorowski** [2]  **Anthony C. Gordon** [2]  **A. Aldo Faisal** [1 3 4 5]

## Abstract

Explainable AI (XAI) is seen as important for AI-driven clinical decision support tools but most XAI has been evaluated on non-expert populations for proxy tasks and in low-fidelity settings. The rise of generative AI and the potential safety risk of hallucinatory AI suggestions causing patient harm has once again highlighted the question of whether XAI can act as a safety mitigation mechanism. We studied intensive care doctors in a high-fidelity simulation suite with eye-tracking glasses on a prescription dosing task to better understand their interaction dynamics with XAI (for both intentionally safe and unsafe (i.e. hallucinatory) AI suggestions). We show that it is feasible to perform eye-tracking and that the attention devoted to any of 4 types of XAI does not differ between safe and unsafe AI suggestions. This calls into question the utility of XAI as a mitigation against patient harm from clinicians erroneously following poor quality AI advice.

## 1. Introduction

In the healthcare landscape, artificial intelligence (AI) is primarily anticipated to manifest within the confines of a clinical decision support system (CDSS) as opposed to functioning as an independent entity, at least in the foreseeable future (Festor et al., 2021). Consequently, fine-tuning the engagement between healthcare providers and the AI-CDSS is imperative for large-scale acceptance and impact, an aspect that has thus far neglected (van de Sande et al., 2021). Explainable AI (XAI) has been put forth as a possible solution by presenting clear and comprehensible rationale for AI-based suggestions to human users (Barredo Arrieta et al., 2020). Besides enhancing overall trust in AI, XAI has been suggested as a tool for averting the potential threat of unusual or even harmful AI advice being unintentionally acted upon (Jia et al., 2022; Gordon et al., 2019; Antoniadi et al., 2021). This is of increasing concern given the rise of generative AI (predominantly as large language models, LLMs) which have a tendency towards hallucinatory (and therefore if applied in a clinical context, unsafe) advice at times (Lee et al., 2023). However, the evidence as to whether XAI can fulfil this role as a defence against inadvertent following of unsafe (i.e. hallucinatory) AI suggestions remains ambiguous at best (Evans et al., 2022; Jacobs et al., 2021; Ghassemi et al., 2021).

Regarding the practical implementation of clinical XAI, there are few clinical evaluations involving XAI with expert end-users and fewer still in a high-fidelity environment (Schoonderwoerd et al., 2021). Recent data suggests that the correlation between the actual prescribing pattern behaviour of doctors and self-reported XAI utility is much lower than anticipated (Nagendran et al., 2023). Notably, other researchers have pointed out that both self-reports and actual behaviours can only be recorded post-event (Cao & Huang, 2022). As a result, the application of these retrospective metrics as part of a reinforcement learning feedback loop is significantly limited compared to other real-time indicators of clinical attention, such as eye-tracking (Ball & Richardson, 2022; Harston & Faisal, 2022). This technology has been employed extensively in scenarios outside of the hospital to determine the focus of an individual's attention (Auepanwiriyakul et al., 2018; Makrigiorgos et al., 2019; Ranti et al., 2020; Harston et al., 2021). A high-fidelity simulation suites provide a platform to scrutinise XAI in a setting that mirrors actual clinical practice while maintaining standardised experimental conditions (and is thereofe frequently used in medical training (Cato & Murray, 2010; Cook et al., 2011)). Our approach therefore addresses the weaknesses of prior work (non-clinical subjects, proxy tasks, low fidelity settings) by integrating eye-tracking within a high-fidelity environment to more accurately gauge the doctor-XAI interaction dynamic.

*Equal contribution  [1]UKRI Centre for Doctoral Training in AI for Healthcare, Imperial College London, UK [2]Department of Surgery and Cancer, Imperial College London, UK [3]Department of Computing, Imperial College London, UK [4]Department of Bioengineering, Imperial College London, UK [5]Institute of Artificial Human Intelligence, University of Bayreuth, Germany. Correspondence to: A. Aldo Faisal <aldo.faisal@imperial.ac.uk>.

*Workshop on Interpretable ML in Healthcare at International Conference on Machine Learning (ICML)*, Honolulu, Hawaii, USA. 2023. Copyright 2023 by the author(s).

In this study, we examined the impact of four different AI explanation types on clinicians within a high-fidelity simulation suite while they carried out a common hospital task: deciding how much of a given drug to prescribe to a patient after evaluating them. We aimed to quantify the effect of XAI on doctors' prescribing decisions and looked specifically to explore whether the attentional engagement of doctors (as measured by eye-tracking) varied between safe and unsafe AI scenarios.

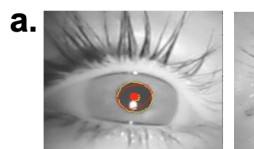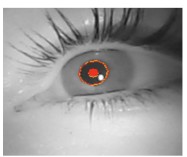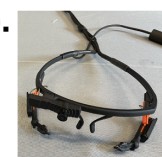

*Figure 2.* **Pupil detection (a) and eye-tracking glasses (b) –** Automatic pupil detection -¿ triangulates gaze position after calibrating software for each subject. Eye-tracking glasses have 3 cameras ('ego-centric' world-view camera plus one camera for each eye).

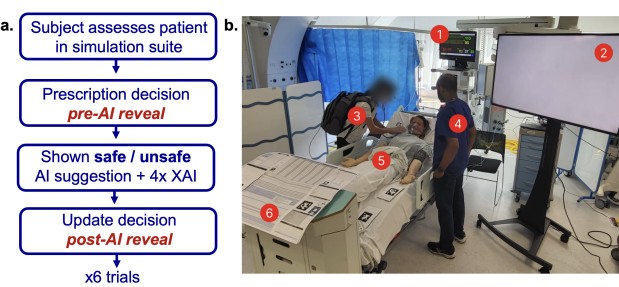

*Figure 1.* **Trial protocol (a) and simulation suite (b) –** where numbers in the simulation suite refer to (1) vital signs monitor, (2) AI screen, (3) subject, (4) bedside nurse (played by experimenter), (5) high-fidelity patient mannequin, (6) bedside ICU data chart

## 2. Methods

**Experimental Setup and AI-driven decision support tool –** Our study entailed an observational exploration of human-AI interaction within a simulation suite. Doctors encountered one of six patient scenarios under either of two conditions: a safe AI recommendation or an unsafe (i.e. potentially hallucinatory) one. The distinction between safe and unsafe was determined by significantly high or low prescriptions of fluid and vasopressors, as established in previous research (Festor et al., 2022). The AI advice given was artificially generated, as the study's main focus was to evaluate the interaction dynamics between medical professionals and AI. We devised four distinct explanations for the simulated AI system, each based on methods we have applied to reinforcement learning decision support systems. The first provided a natural language narrative of the Q-value difference between the suggested action and other potential actions. The second elucidated the predicted short-term mortality changes following dosing alterations as forecast by the AI. The third highlighted the five most impactful features of the input data that influenced the AI's suggestion. Lastly, the three most influential training examples during the Q-learning process were identified.

**Eye-tracking for Gaze Detection –** We made use of eye-tracking for gaze detection as a surrogate marker to ascertain the focus of clinicians' attention during simulations and its variability. Participating subejcts wore non-obtrusive,

commercially available eye-tracking glasses (Pupil Labs Core) equipped with three cameras (Figure 2b). The primary camera recorded the subject's viewpoint, while the other two focused on their eyes. The Pupil Labs software (Pupil Capture, version 3.5.7) used the eye cameras to outline the pupil and determine where the gaze was directed within the world-view (Figure 3a).

A 2D calibration exercise was conducted before the experiment, which consisted of two stages. Initially, a static calibration exercise was done with five screen markers on a laptop screen (default Pupil Labs 'screen marker' calibration). Subsequently, a depth-oriented static exercise was executed, with doctors sequentially concentrating on nine screen markers ('natural features' mode) on a 60-inch TV screen, first at 1 metre and then at 2 metres distance. The depth variation aided calibration for real-world environments where natural head movements were expected. The eye-tracking glasses were linked to a laptop (Lenovo Thinkpad) throughout the experiment, housed in a lightweight backpack worn by participants for unrestricted movement within the suite.

We defined four key regions of interest (ROIs) (Figure 1b): the patient mannequin (Simman 3G, Laerdal Medical, Stavanger, Norway), the vital signs monitor, the paper intensive care unit (ICU) data chart, and the AI display screen. Within the AI display screen, we further identified four sub-regions corresponding to the four types of AI explanation. ROIs were determined in the post-processing phase by recognizing pre-placed QR codes (known as April tags, refer Figure 3a) within the simulation suite to define ROI boundary boxes. Post-processing allowed for the analysis of the following eye-tracking metrics: (i) gaze-time per ROI, (ii) number of fixations per ROI (a fixation is the most common eye movement, occurring when eyes halt scanning and focus the foveal area of the visual field in one location), (iii) blink rate (per minute) per ROI. Blink rate is typically inversely proportional to concentration or focus on an object.

We also developed a unique proxy for behavioural attention, which takes into account the portion of the visual field occupied by a region of interest (ROI). In essence, if an ROI takes up 50% of the visual field for half of the time, we

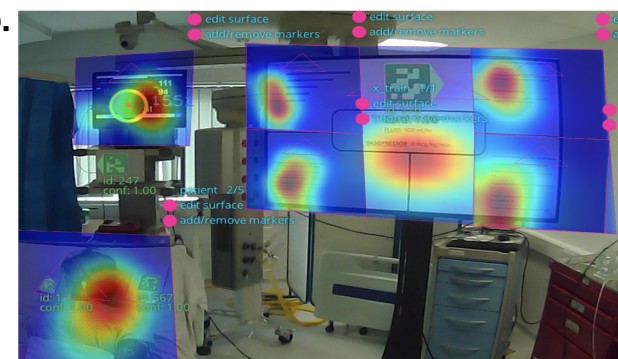

*Figure 3.* **Post-processing of eye-tracking data -** Left-hand image shows bounding boxes around regions (surfaces) of interest (ROIs). Right-hand side shows gaze density per ROI as heatmaps.

would expect, based purely on randomness, that the gaze would fall within the ROI 25% of the time during the experiment. Comparing this 25% 'random gaze' figure with the actual gaze proportion allows us to compute a proxy for the relative significance of ROIs by juxtaposing the proportions of random and actual gaze.

**Simulation Experiment –** Subjects initially received a standard experiment briefing. This was followed by a pre-experiment questionnaire on attitudes towards AI and demographics. Following orientation in the simulation suite and eye-tracking calibration exercises, they commenced the simulated scenarios. An experimenter role-played the ICU bedside nurse. The doctors were required to assess six simulated ICU patients with sepsis.

Within each of the six scenarios, clinicians were tasked with conducting an assessment, which included reviewing available patient data and performing patient examinations. Following this, the bedside nurse requested their prescription for fluid and vasopressors for the subsequent hour of the patient's admission, their confidence level in their prescription, and whether they would seek senior advice (or a second opinion in the case of a consultant). Doctors were then presented with the AI suggestions and explanations on a large display near the patient bed, after which they were asked to confirm or modify their prescription doses and revise their responses to the confidence and senior advice questions (Figure 1a).

**Subject recruitment –** Recruitment of ICU doctors used targeted advertising and convenience sampling to a local hospital region. Inclusion criteria were: (i) practising doctor, (ii) has worked for two or more months in an adult ICU, (iii) currently works in ICU or has worked in ICU within the last 6 months. Subjects went through a standard informed consent procedure prior to taking part and were compensated for their time. Each experiment lasted approximately 60

minutes. The study received ethical approval from the Research Governance and Integrity Team (RGIT) at Imperial College London and the Health Research Authority (Ref: 22/HRA/1610).

## 3. Results

**Cohort recruited –** So far, twelve ICU doctors with eye-tracking data available were included (8 male, 4 female). Mean doctor age was 31 years (standard deviation (SD) 5 years). Mean ICU experience was 2.5 years (SD 3.2 years, range 2 months to 10 years).

**Eye-tracking metrics on ROIs –** There were more gaze fixations for the AI suggestion during unsafe scenarios but this was only significant for ICU doctors (¿1 year experience in ICU, p=0.046, Figure 4). There were no significant differences in number of gaze fixations between the different XAI modalities for either safe or unsafe scenario, regardless of doctor seniority.

Mean blink rate was lowest for the ICU chart (5.7 blinks per minute (bpm), SD 4.4), similar for both patient and vital signs monitor (mean 15.0 bpm and 14.9 bpm, SD 9.2 and 9.4 respectively) and notably higher for the AI screen (mean 18.4 bpm, SD 7.6). When comparing all conventional clinical ROIs (chart, patient, monitor; blue bars in Figure 5) to all AI ROIs (including XAIs; orange bars in Figure 3), there was a significantly lower mean blink rate on the conventional clinical ROIs than the AI ROIs (11.5 bpm vs. 22.7 bpm, p=0.005).

For every ROI except the patient mannequin, there was a significantly higher actual gaze proportion than random chance gaze proportion (p¡0.001 for all seven comparisons). For the major ROIs (AI screen, ICU chart, vital signs monitor, patient) the ratio of actual gaze to random gaze was 6.7, 1.6, 11.8 and 1.2 respectively. For the XAI ROIs (training

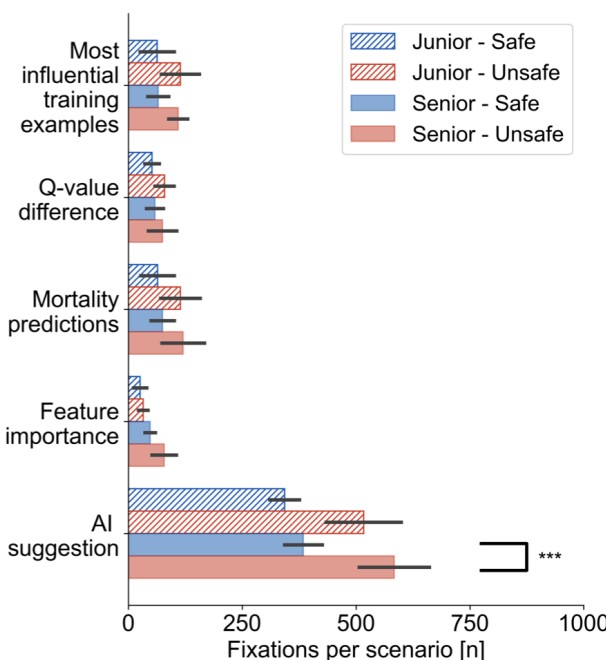

*Figure 4.* **Fixations per scenario by safety status of scenario and experience level of doctor –** Mean and SEM error bars.

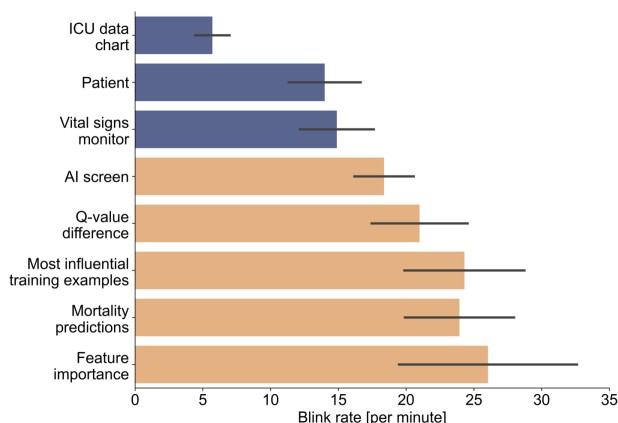

*Figure 5.* **Blink rate by region of interest –** Mean and SEM error bars. Blue bars are traditional clinical surfaces while orange bars are AI / XAI surfaces.

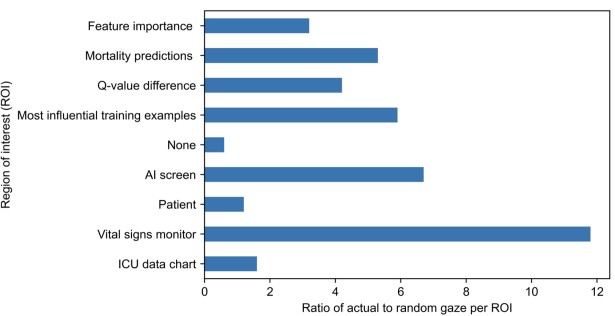

*Figure 6.* **Ratio of actual to random gaze per ROI –** Random gaze is the proportion expected based on the area occupied by an ROI within the visual field (i.e. if ROI takes up 50% of the screen, 50% of the time, we would expect gaze to randomly fall within it 25% of the time). Actual gaze is the proportion observed during the experiment.

examples, Q-value difference, mortality, feature importance) the ratio of actual gaze to random gaze was 5.9, 4.2, 5.3 and 3.2 respectively (see Figure 6).

**Self-reported XAI usefulness –** Self-reported data on the utility of XAI was only available for 6 of the 12 subjects (Figure 7). The overall mean post-experiment usefulness rating for the XAI was 3.5 (SD 0.8) on a zero to four scale with higher value implying the XAI was more useful. The training examples explanation was the only one of the four to be rated significantly lower than the overall rating for explanations in general (mean 1.0, SD 0.9, p¡0.001). When comparing the 'objective' marker of how many fixations there were on the four different types of XAI to the 'subjective' marker of how clinicians rated the usefulness of the four XAIs, we found no significant correlation for any of the four XAIs.

**Adherence to AI suggestions among doctors –** We defined adherence to AI as the distance between a doctor's final prescription (having had the opportunity to view the AI suggestion) and the value of the AI suggestion for any given trial/scenario (higher distance suggesting that the doctor was less adherent to AI and vice versa). For fluid, there was no significant difference in adherence between safe and unsafe AI suggestions (absolute difference of 208 ml/hr and 167 ml/h distance respectively between doctor and AI, p=0.47). For vasopressor, there was significantly more ad-

herence for safe AI than unsafe AI (0.05 mcg/kg/min and 0.30 mcg/kg/min distance respectively between doctor and AI, p¡0.0001). However, there was no evidence of correlation between eye-tracking metrics (blink rate or number of gaze fixations) and AI adherence regardless of safety status or drug. Nor was there evidence of correlation between number of gaze fixations and AI adherence regardless of safety status or drug. Nor was there a significant association between number of fixations specifically on XAI ROIs and drug (either fluid or vasopressor) for either AI scenario (safe or unsafe).

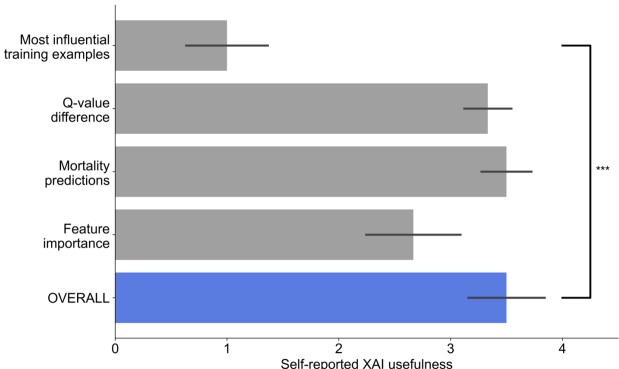

*Figure 7.* **Usefulness rating for each type of explanation as well as overall rating –** Mean and SEM error bars. The only significantly different explanation type compared to overall was the 'most influential training examples'.

## 4. Discussion

Our study reveals several new findings that augment our knowledge base on clinicians' interaction with AI-assisted decision support instruments and their associated explanations. First, the utility of gaze fixations and blink rate as surrogate indicators for attention to an AI tool was successfully demonstrated within a high-fidelity simulation setup. The extension of this to real-life clinical situations would be contingent on the availability of less intrusive eye-tracking hardware and addressing privacy concerns linked to video recording of staff and patients. Second, we observed that there was no marked elevation in attention to any of the four explanation types when dealing with an unsafe AI suggestion versus a safe one. This negates the assumption of increased reliance on explanations during unsafe scenarios. Third, there was no correlation between the self-reported utility of explanations and the degree of attention they received, suggesting that self-reports alone are insufficient for assessing XAI tools. Fourth, blink rate, a potential marker of cognitive effort, was lowest for the ICU chart packed with granular data and highest for the AI screen and explanations, suggesting less cognitive effort is required to interpret the AI data. Fifth, we were unable to draw consistent connections between eye-tracking metrics and clinical practice variations or adherence to AI suggestions.

Several limitations must temper these findings. First, despite the high fidelity of our simulation suite, it falls short of replicating the intricate dynamics of a real hospital environment, such as dynamic patient examination, continuous patient observation, and multi-disciplinary team interactions before considering an AI recommendation. However, these complexities make it challenging to standardise real-world experiments and require unfeasibly large sample sizes. Hence, simulation experiments remain vital for exploring human-AI

interaction dynamics before embarking on larger real-world studies. Second, our small sample size implies that certain comparisons could have been significantly different with a larger number of participants. Third, the categorisation of AI suggestions into safe or unsafe inherently imposes an arbitrary boundary on a continuous spectrum. Fourth, the explanations varied in their presentation format: some were primarily graphical, and others were text-heavy, potentially confounding comparisons between explanations. Fifth, the safe and unsafe AI suggestions were synthetic and this may not accurately capture how a real AI system might hallucinate an incorrect explanation. As an example, LLMs can often be confidently wrong. This may be further exacerbated during reinforcement learning with human feedback (RLHF) which optimises the output towards human preference and thereby potentially makes wrong answers even more convincing.

Regardless of these constraints, our findings offer invaluable insights into the optimisation of XAI-based medical decision support tools when assessed alongside existing literature. One prevalent presumption is that explanations should aid users in correctly discarding poor or unsafe AI advice. This process typically involves the following steps: [A] an unsafe AI suggestion is presented to the user, [B] an explanation for the AI suggestion is subsequently offered, [C] the user either identifies a flaw in the explanation or cannot find reasons justifying the inappropriateness of the AI suggestion, and finally [D] the user discards the unsafe AI advice. Evidence from a study in non-clinicians hints at a possible breakdown in the causal link between steps [C] and [D] (Shafti et al., 2022). The authors discovered that the presence of an explanation amplified the impact of AI advice, with the quality of explanation seemingly irrelevant, indicating a potential automation bias where the mere existence of an explanation is used as a heuristic to follow the AI advice, bypassing critical evaluation. The risk of such automation bias is well documented in other clinical investigations (Micocci et al., 2021; Panigutti et al., 2022).

Another piece of evidence is an experiment examining a mental health drug decision support tool, where explanations failed to prevent clinical users from following intentionally subpar AI recommendations (Jacobs et al., 2021). Our study corroborates that the substantially higher rejection rate of unsafe advice over safe advice was not driven by an increased reliance on, or attention paid to, explanations. We validated this by concurrently evaluating the trio of doctors' actual prescription decisions, their visual attention during decision-making, and their post-experiment subjective explanation ratings.

The deployment of eye-tracking technology in AI-user studies has been minimal to date, with one significant example being the work by Cao and colleagues, who utilized a spatial

reasoning task and detected a positive association between gaze percentage on the AI suggestion and both perceived user reliance on AI and their agreement with AI suggestions (Cao & Huang, 2022). Interestingly, gaze did not appear to correspond with perceived trust. In line with this, our study also found no correlation between subjective explanation ratings and the number of gaze fixations on the AI explanation. One of the touted benefits of using eye-tracking for assessing human-AI interactions is its real-time nature, which contrasts with retrospective subjective ratings or human agreement with AI suggestions, potentially forming the backbone of an adaptive collaboration feedback loop (Cao & Huang, 2022). Our results caution that before eye-tracking can take center stage in such a feedback system involving XAI, there is a pressing need to establish robust eye movement patterns that can accurately categorize users and, optimally, forecast their interactions with an AI system (though this might be possible with a larger sample size).

## 5. Conclusion

Overall, our findings suggest that eye-tracking is a feasible method for evaluating clinicians' interactions with XAI. We demonstrate that clinicians' responses to safe and unsafe AI are noticeably different. However, the lack of 'rescue' effect provided by XAI raises questions about its role in preventing patient harm from clinicians following poor quality AI advice. Our findings underscore the need for the next generation of AI decision support tools to tailor not only their advice but also the manner in which they interact with, and provide explanations to, their clinician end-users.

## Acknowledgements

MN and PF are supported by the UKRI CDT in AI for Healthcare (EP/S023283/1). ACG is supported by an NIHR Research Professorship (RP-2015-06-018). AAF is supported by a UKRI Turing AI Fellowship (EP/V025449/1). This work was funded by the University of York and the Lloyd's Register Foundation through the Assuring Autonomy International Programme (Project Reference 03/19/07) and supported by the National Institute for Health Research (NIHR) Imperial Biomedical Research Centre (BRC). The views expressed in this publication are those of the author(s) and not necessarily those of the NIHR or the Department of Health and Social Care.

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

## A. Appendix - Standardised experiment briefing to participants

This simulation experiment aims at studying how clinicians might interact with an AI decision support tool for patients with sepsis. The AI takes patient variables as input and outputs a treatment recommendation for both fluid and vasopressor. We have some evidence of effectiveness of the AI you will interact with on retrospective data from an American dataset and further validation on retrospective data from the Netherlands. However, there has not yet been prospective evidence of either effectiveness or safety.

You will conduct a brief ward round review of 6 ICU patients with sepsis. Between each patient, you will exit the room and be called in again to see the next patient by the nurse.

For each patient, you will first be asked:

- Treatment prescriptions for:

- Fluid in ml for the next hour

- Vasopressor in mcg/kg/min for the next hour

- Your confidence in the prescription on a scale from 1 (low) to 10 (high)

- Whether or not you want to get advice from another doctor / senior doctor

You will then be shown the AI treatment recommendation on a digital screen. The screen contains the AI dose suggestions in the middle (for fluid and vasopressor) along with 4 explanations for the suggestions (one in each corner)

You will then be asked:

- To what extent you agree with the AI suggestion on a scale from 1 (strongly disagree) to 5 (strongly agree)

- Whether you wish to adjust your treatment prescription for:
    - Fluid in ml for the next hour
    - Vasopressor in mcg/kg/min for the next hour

- Whether your confidence in your prescription (on a scale from 1 [low] to 10 [high]) has changed as a result of seeing the AI suggestion

- Whether or not you want to get advice from another doctor / senior doctor

- If the AI suggested treatment was to be administered to the patient, would you act to stop the administration?

We will now show you an example of the AI screen that you will encounter in the experiment. You can see that there is an AI suggestion in the centre of the screen which is how much fluid and vasopressor the AI recommends over the next hour.

Around the corners of the screen there are four different types of explanation which can be thought of as the AI trying to convey the rationale for its suggested doses. The mortality change explanation conveys information on what the AI predicts will happen to the overall mortality risk in the short-term based on potential dose increases or decreases. The most influential training examples explanation conveys which three training cases were most helpful for learning the current suggestion in the same way that we might base our own treatment choices on previous notable cases we learnt from. The feature importance explanation conveys which were the top five features (or items in the data) that were most useful to the AI in generating its current suggestion. The 'AI treatment options gap' explanation conveys how much one treatment strategy looks superior compared to alternatives. If all potential options are similar (i.e. a low gap) then it suggests that the AI has near equipoise for options and you may wish to use your own judgement more strongly (as you will have additional information from examining the patient for example). However, if the gap is high it suggests that the AI has identified one particular treatment strategy as superior to the alternatives and therefore it would be worth considering this recommendation more strongly than with a low gap recommendation.

## B. Appendix - AI explanations used

**Q-value difference –** This approach leverages the fact that once training is complete, the reinforcement learning (RL) Q table will contain Q values for any given state-action pair. The optimal action is the one with the highest Q value for any given state. However, the difference between this highest Q value and the alternatives might be very small or large. If large, then there is much higher anticipated value from following the recommended action compared to an alternative. This is dichotomised arbitrarily from a continuous value to make interpretation more simple for non-AI users.

**Mortality predictions –** This approach leverages the fact that mortality can be predicted for any given state in the RL state space. Therefore the impact of different dosing strategies that might result in transition to alternative states with different predicted mortalities can be displayed to the subject to highlight how alternate strategies might change the risk of death.

**Feature importance –** This approach leverages the fact that the state space for RL based sepsis algorithms is commonly constructed using a k-means clustering algorithm to enable dimensionality reduction. After the algorithm converges, the cluster centroids represent the average feature values for patients in a particular state/cluster. A new patient would be assigned to the state/cluster that minimised the distance from their feature values to the respective cluster centroid. Intuitively, with often over 40 features, some features will be closer to the cluster centroid value than others for any patient assigned to a given state. This is exploited to rank features in terms of their proximity to the cluster centroid (or average state feature values) given that the archetypal patient for whom an RL agent policy action most applies is a patient who is most typical of that state. So subjects can be shown the top five ranked features contributing to state assignment.

**Most influential training examples –** This approach leverages the fact that the difference in Q values for any given state-action pair between one iteration of Q-learning and the previous iteration reflects how valuable the currently seen training episode is for learning the optimal action for any given state. This is similar to an instance-based explanation used in deep learning imaging XAI where the explanation consists of showing similar image instances from the training instance to explain why a particular image classification has been made.

- Create empty Q and 'Q-difference' tables (both indexed by Q(S,A) tuple)

- For 500,000 episodes:
    - Select random training episode:
        * For each time-step:
            · Perform Q learning
            · Check the Q differences table → is the Q difference (i.e. difference between old and updated new Q values) from this step among the top 3 for this state-action tuple?
            · If so → update differences table with the ID for this episode

- End with a dictionary of top 3 influential episodes per Q(S,A) tuple

