# OpenReview forum: "Eye-tracking of clinician behaviour with explainable AI decision support: a high-fidelity simulation study"
_ICML.cc/2023/Workshop/IMLH — IMLH 2023 Oral_

### Official Review · Reviewer_brBq · 2023-06-16
**Eye-tracking is feasible to evaluate the clinicals' interactions with XAI.**

**Rating:** 7
**Confidence:** 2

**Review:**

Real nice work, this paper provides a new approach to evaluate XAI's suggestions in a high-fidelity environment. Even though I'm not an expert in XAI, and, obviously (as mentioned in the discussion) a direct experiment in a real hospital environment would be of interest, the insight of this short paper is already convincing enough.

---

### Official Review · Reviewer_y6uE · 2023-06-18
**Well-motivated empirical study with interesting insights and implications**

**Rating:** 7
**Confidence:** 4

**Review:**

This paper presents a high-fidelity simulation study involving intensive care doctors to investigate the effectiveness of eXplainable AI (XAI) methods under safe and unsafe AI behavior. The study addresses the crucial need to evaluate XAI systems in realistic settings with expert users, deviating from the common practice (due to well-known, understandable reasons) of using non-expert populations for proxy tasks and in low-fidelity settings.

The study employs eye-tracking technology and derived metrics to examine the utility of various XAI methods. The empirical evaluation demonstrates that gaze fixations and blink rate can serve as reliable surrogate indicators for attention within the simulated environment. Surprisingly, there is no significant increase in attention observed for explanations during unsafe scenarios, challenging the assumption of increased reliance on explanations in such situations. This discrepancy is noteworthy as it contradicts the self-reported usefulness of explanations, which fails to align with the actual attention received. Consequently, the results underscore the importance of employing objective assessment methods when evaluating XAI tools and raise concerns about the effectiveness of XAI as a strategy to mitigate potential patient harm resulting from clinicians mistakenly following low-quality AI recommendations.

Despite acknowledging the study's limitations, which the authors have transparently addressed, this research represents a promising step towards evaluating and analyzing the practical utility of XAI methods. It provides valuable and sometimes counter-intuitive insights, contributing to the advancement of the field.

---

### Official Review · Reviewer_ArMg · 2023-06-18

**Rating:** 6
**Confidence:** 3

**Review:**

The paper addresses the question of whether explainable AI (XAI) can serve as a safety mitigation mechanism in healthcare settings. The authors highlight the importance of studying the interaction between healthcare providers and XAI systems, particularly in high-stakes scenarios such as clinical decision-making. They conducted a study with intensive care doctors using eye-tracking glasses to analyze their engagement with XAI suggestions, both safe and unsafe (i.e., hallucinatory). The findings indicate that the attention devoted to XAI did not differ significantly between safe and unsafe scenarios, raising doubts about the utility of XAI as a defense against following poor-quality AI advice and potential patient harm. The paper presents a thorough examination of XAI's role in mitigating safety risks, utilizing high-fidelity simulation and eye-tracking technology to provide valuable insights into the dynamics of doctor-XAI interaction.

The study contributes to the understanding of XAI's practical implementation in healthcare by incorporating expert end-users and a high-fidelity environment, which adds credibility to the findings. By employing eye-tracking technology, the authors overcome some limitations of relying solely on self-reports or post-event recordings, allowing for real-time assessment of clinical attention. The use of a high-fidelity simulation suite mirrors actual clinical practice and provides standardized experimental conditions. However, it is important to note that the paper acknowledges the ambiguity surrounding the effectiveness of XAI in mitigating inadvertent following of unsafe AI suggestions. Further research and evaluation are needed to fully ascertain the potential of XAI as a safety mechanism in clinical decision support systems. Overall, this study contributes to the ongoing discussions on the practical application and utility of XAI in healthcare, shedding light on the interaction dynamics between intensive care doctors and XAI systems in a realistic setting.

---

### Official Review · Reviewer_6pFA · 2023-06-19
**Review of "Eye-tracking of clinician behaviour with explainable AI decision support"**

**Rating:** 9
**Confidence:** 4

**Review:**

__Overview__: the authors of this long paper designed a user study to examine the value of eye-tracking as a tool for evaluating how clinicians utilize recommendations from AI systems. There are limited studies examining this.

__Positives__:
1. Well written paper that was easy to follow and enjoyable to read
2. Well designed and high-quality user study which provides insights that will be valuable to the research community
3. Provides evidence which corroborates the idea that clincians may not rely on explanations of AI recommendations to reject unsafe recommendations, questioning the utility of XAI.

__Negatives__:
1. Difficult to reproduce + small sample size
2. Given that the provided unsafe suggestions were synthetic, this may not accurately capture how a real AI system may hallucinate an incorrect explanation. For example, LLMs like ChatGPT are often confidently wrong. However, being trained with RLHF optimizes the output towards human preference, which may make wrong answers more convincing.
3. Difficult to evaluate how novel XAI methods might perform without reproducing the study

---

### Meta-Review · Area_Chair_qtg5 · 2023-06-19

**Recommendation:** Accept (Oral)
**Confidence:** 5

**Metareview:**

The paper provides a high-fidelity simulation study in clinical settings on eye-tracking behavior of clinicians with XAI.

Pros:

The study is timely, the design is solid, and the conclusion is insightful for the XAI in healthcare community.


Cons:

No ethics approval or consenting process were mentioned. The authors should address these points in the revised manuscript.

(Good to have) In addition to the quantitative data reported, it would be great to collect some qualitative data about doctors' comments on their perception and usefulness of the AI and explanations. So that the rich details of users' subjective perception can be compared with the objective perception measure using eye-tracking.

I recommend this paper and suggest the authors addressing the reviewers comments in their revision.

---

### Decision · Program_Chairs · 2023-06-20

Accept (Oral)